# Parasitosis in Pet Dogs from Rondônia, Amazon Biome, and Human Perception of Zoonoses

**DOI:** 10.3390/ijerph21020138

**Published:** 2024-01-26

**Authors:** Talita Oliveira Mendonça, Patricia Parreira Perin, Dayane da Silva Zanini, Hortência Laporti de Souza, Paulo Henrique Kanopp Pires, Igor Mansur Muniz, José Hairton Tebaldi, Luís Antonio Mathias, Karina Paes Bürger, Estevam G. Lux-Hoppe

**Affiliations:** 1Parasitic Diseases Laboratory (LabEPar), Departament of Pathology, Reproduction, and One Health (DPRSU), School of Agricultural and Veterinary Studies (FCAV), São Paulo State University (UNESP), Jaboticabal 14884-900, SP, Brazil; talita.mendonca@unesp.br (T.O.M.); patricia.perin@unesp.br (P.P.P.); jose.h.tebaldi@unesp.br (J.H.T.); la.mathias@unesp.br (L.A.M.); karina.burger@unesp.br (K.P.B.); 2School of Veterinary Medicine and Animal Science (FMVZ), São Paulo State University (UNESP), Botucatu 18618-681, SP, Brazil; d.zanini@unesp.br; 3Department of Veterinary Medicine, Federal University of Rondônia (UNIR), Porto Velho 76801-974, RO, Brazil; tensalaporti@gmail.com (H.L.d.S.); renopp.representacoes@gmail.com (P.H.K.P.); igor.mansur@unir.br (I.M.M.)

**Keywords:** endoparasites, ectoparasites, one health, zoonoses

## Abstract

Zoonotic parasitic diseases in dogs are particularly concerning in regions with low human development indices due to inadequate sanitary services and insufficient environmental and health education. This study aimed to assess the parasitological status of dogs living in households and evaluate their owners’ knowledge about zoonoses. A total of 183 dogs from Rolim de Moura, Rondônia State, were screened for the presence of ectoparasites, and 163 fecal samples were collected for analysis. The results showed that 74.23% (112/163) of the animals had at least one species of endoparasite. The most identified pathogens were *Ancylostoma* spp. (68.71%, 112/163), *Trichuris vulpis* (11.66%, 19/163), *Toxocara canis* (6.75%, 11/163), *Cystoisospora canis* (4.91%, 8/163), *Dipylidium caninum* (1.23%, 2/163), and *Hammondia/Neospora* (0.61%, 1/163). Ectoparasites were observed in 43.17% (79/183) of the evaluated animals, with *Rhipicephalus sanguineus* found in 31.15% (57/183) and *Ctenocephalides felis felis* in 20.77% (38/183). Only 11.48% (7/61) of the owners were familiar with the term “Zoonoses.” However, a significant majority (83.61%, 51/61) believed that dogs can transmit diseases to humans. Our findings highlight the prevalence of parasites in the studied area and associated risk factors, underscoring the urgent need for educational interventions to raise awareness about these diseases and their risks to human health.

## 1. Introduction

Parasitism in pet dogs is relatively common, and the proximity between these animals and humans may facilitate the transmission of zoonotic parasites. Various biotic and abiotic factors, such as animal behavior, management practices, housing conditions, and parasite control strategies, were shown to influence the dynamics of parasitic infections in dogs [1,2]. However, parasite control is often neglected by owners due to a lack of knowledge about parasitic diseases in dogs and their risks to human health [3]. Therefore, it is crucial to raise awareness and provide education on these topics to promote effective preventive measures and safeguard the health of both animals and humans. Considering the social and economic impacts of these diseases, the adoption of preventive measures becomes necessary [4,5]. 

Zoonotic parasitic infections have been described in dogs worldwide [1], but they are more frequent in regions with low human development [6]. These regions often face challenges related to a lack of basic sanitation, limited access to healthcare, and inadequate public health infrastructure. In Brazil, the northern, central-western, and northeastern regions of the country, which have low human development indices, record the highest rates of infectious and parasitic diseases in humans [5]. 

The northern region of Brazil, predominantly located within the Amazon Biome, has experienced relatively recent human occupation, with significant expansion starting in the 1970s. This region is renowned for its biodiversity but faces substantial anthropogenic pressure due to extractive activities and agricultural potential [7]. Gastrointestinal parasitosis in dogs is highly prevalent in this region, exceeding 80% in certain areas [2,8,9,10]. However, there is a dearth of data regarding pet dog owners’ perceptions of parasitic diseases in dogs and their significance as zoonoses in this region, despite the existence of perception studies conducted in other areas of Brazil and globally [3,11,12,13]. Consequently, further studies in unexplored areas are warranted, as risk perception is known to vary across countries and regions due to diverse factors, including cultural, environmental, and governmental influences [1,11].

The objective of this research was to identify the occurrence of parasitosis in pet dogs in a municipality in the state of Rondônia and investigate the population’s perception of parasites and zoonoses.

## 2. Materials and Methods

### 2.1. Study Area

The study was conducted in the Rolim de Moura municipality, situated at coordinates 11°48′13″ S and 61°48′12″ W, within the state of Rondônia in the northern region of Brazil. This municipality is located in the Zona da Mata region, which is part of the Amazon Biome, at an elevation of 290 m above sea level. Rolim de Moura covers a total area of 1457.888 km^2^ (Figure 1). The Anta Atirada River hydrographic basin is found within Rolim de Moura. The region features a tropical climate with a distinct dry season (Aw) that is characterized by an annual temperature range of 19 °C to 36 °C and annual precipitation ranging between 1700 and 1900 mm. Rolim de Moura has a Human Development Index of 0.700, ranking seventh in Rondônia and 1904th among the 5568 Brazilian municipalities. As of 2021, the population of the municipality stood at 55,748 residents, occupying an area of 1457.81 km^2^. This translates to a population density of 34.74 inhabitants per km^2^ [14,15].

The study focused on 12 distinct sectors within the municipality: Beira Rio, Boa Esperança, Centenário, Centro, Cidade Alta, Industrial, Jardim Eldorado, Jardim Tropical, Nova Morada, Olímpico, Planalto, and São Cristóvão (Figure 2).

The inclusion criteria for this study was the consent of dog owners, given the absence of official data on the actual canine population size in the municipality. A total of 183 dogs from 61 households within the study area participated (see Figure 3 and Appendix A).

Initially, each animal underwent a comprehensive physical examination, coupled with a meticulous inspection for ectoparasite presence. Given the observed infestation in all animals, ectoparasite samples (28 in total) were collected on a per-household basis and subsequently preserved using 70% ethanol in appropriately labeled containers. The taxonomic identification of ectoparasites was accomplished using specific keys for each arthropod group [16,17,18,19,20,21].

Fecal samples were obtained through enema or immediate defecation. The 163 samples collected were stored in temperature-controlled containers during transportation and processed within 24 h of collection. To diagnose endoparasitic presence, the techniques outlined by Gordon and Whitlock, Willis, Watanabe et al., and Baermann [22] were employed. Descriptor calculations for endoparasitic infection were conducted using the methodology described by Bush et al. [23].

### 2.2. Evaluation of Owners’ Knowledge and Animal/Environmental Characteristics 

A semi-structured interview based on the provided script (Appendix B) was conducted to gather information on the owners’ understanding of canine diseases and zoonoses, as well as details about the animals, methods for parasitic control, and sanitation practices. This questionnaire was administered upon completion of the sample collection.

### 2.3. Statistical Analyses

The associations between parasite infection/infestation and host and environmental characteristics were examined through univariate and, when applicable, multivariate regression analyses. The explanatory variables encompassed sex, size, breed, age, outdoor access, the use of anthelmintics, employment of ectoparasiticides, residence location in relation to the river, street paving, sewage system, water source, frequency of veterinary consultations, and geographic location. The outcome variables comprised overall gastrointestinal parasite infection, overall ectoparasite infestation, infection by *Ancylostoma* spp., infection by *Toxocara canis*, infection by *Trichuris vulpis*, infection by *Cystoisospora canis*, infection by *Dipylidium caninum*, infestation by fleas, and infestation by ticks, as well as coinfections involving *Ancylostoma* and *Toxocara*, *Ancylostoma* and *Trichuris*, and *Ancylostoma* and *Cystoisospora*.

With respect to the dog owners, we assessed their educational level as an explanatory variable and knowledge regarding zoonotic diseases, as well as the potential for dogs to transmit diseases to humans, as outcome variables. Additionally, we evaluated environmental factors (such as water source and sewage system) as explanatory variables and the presence or absence of animals infected/infested by any parasite as outcome variables. 

The chi-square test or Fisher’s exact test was utilized to assess associations, considering *p*-values below 0.2 as indicating significance as screening. Factors demonstrating significant associations underwent further scrutiny through simple logistic regression analysis, with a *p*-value threshold set at less than 0.05. The statistical analyses were conducted using Epi Info software (version 7.2.3.1, 2019). Variables displaying a significant association (*p* < 0.05) were then subjected to the Akaike information criterion to identify the optimal model by employing R software (version 4.3.2). Subsequently, a mixed model was developed, considering the significant associations, using the lme4 package in R software. The confidence interval for prevalence was calculated using the exact method implemented in the “binom” package of R software.

## 3. Results

The analysis of 163 fecal samples revealed that 74.23% (121/163) tested positive for at least one species of endoparasite (Table 1 and Figure 4). The quantification of eggs per gram of feces ranged from 50 to 46,100 eggs. Among the identified parasites, *Ancylostoma* spp. displayed the highest abundance, with egg counts spanning from 50 to 46,100 eggs per gram. Following *Ancylostoma* spp., *Toxocara canis* exhibited egg counts ranging from 200 to 4200 eggs per gram, while *Trichuris vulpis* demonstrated counts ranging between 50 and 750 eggs per gram. Notably, no lungworm larvae or *Strongyloides stercoralis* were detected using the Baermann test. Ectoparasites were observed in 43.17% (79/183) of the examined animals. Among these, immature and/or adult stages of *Rhipicephalus sanguineus* were found in 31.15% (57/183) of the animals, while *Ctenocephalides felis felis* were identified in 20.77% (38/183). Some animals exhibited co-infestation by both ectoparasite species.

Table 2 provides a detailed breakdown of the descriptors of endoparasitic infection, which were classified according to the municipality sectors. The analysis of these households revealed that 60.66% (37/61) were located on unpaved streets. Given the absence of a sewage system, a significant majority of 90.16% (55/61) relied on septic tanks. Concerning the water supply, 72.13% (44 out of 61) of households accessed water from the public supply, while 37.70% (23/61) relied on artesian wells. Additionally, 9.84% (6/61) had access to water from both sources (further details are available in Table 3 and Table 4).

Observations indicated that co-infection by multiple parasites occurred more frequently than infection by a single parasite. Notably, co-infection involving *Ancylostoma* spp. and *Toxocara canis* was particularly prevalent. The occurrence of polyparasitism was directly associated with contact with other dogs (*p* = 0.04). Furthermore, the large breed dogs exhibited a higher risk of *Ancylostoma* spp. and *Trichuris vulpis* infection (*p* = 0.047, odds ratio (OR) = 2.8352, confidence interval (CI) = 1.0619–8.8003) compared with the small breed animals.

When considered individually, animals positive for *Toxocara canis*, *Trichuris vulpis (*Appendix C and Appendix D), *Cystoisospora canis*, *Dipylidium caninum*, and coinfection with *Ancylostoma* and *Cystoisospora* did not show significant associations with the studied variables.

The analysis revealed a direct association between the extent of contact with other dogs and the frequency of parasitism, particularly with *Ancylostoma* spp. Additionally, the infected animals demonstrated a lower prevalence of parasitic infections when subjected to regular health examinations (*p* = 0.024, OR = 0.80, CI = 0.60–0.95) and had limited or no access to outdoor environments (*p* = 0.020, OR = 0.23, CI = 0.06–0.84) (Table 3). Furthermore, routine veterinary examinations were found to reduce the risk of *Ancylostoma* spp. infection (*p* = 0.02) (refer to Table 4).

The purebred dogs demonstrated significantly lower ectoparasite prevalence in comparison with the mixed breeds (*p* = 0.03, OR = 0.0105, CI = 2.02 × 10^−4^–0.1344), particularly in relation to fleas (*p* = 0.02, OR = 1.91 × 10^−6^, CI = 0.00–6.05 × 10^−3^).

As reported by the dog owners, 76.50% (140/183) of them utilized anthelmintic medications for their dogs. However, among the 61 owners interviewed, 62.84% did not adhere to any specific criteria for usage. Of those who did, 24.04% administered them quarterly, 11.48% semiannually, and 1.64% monthly. The most commonly used products were anthelmintic compounds based on 4H-pyrimidine and pro-benzimidazole, either combined or without a pyrazinoisoquinoline derivative. Among the frequently used drug combinations were pyrantel pamoate + praziquantel + febantel (54.10%), pyrantel pamoate + praziquantel (25.68%), and pyrantel pamoate (8.20%). Approximately 3.83% of the owners were uncertain about the specific anthelmintic used.

Concerning ectoparasiticides, 73.22% (134/183) of the owners employed these products. Among the 61 interviewed owners, 56.83% did not follow any specific criteria for ectoparasiticide use. Among those who did, 22.95% applied them semiannually, 9.84% quarterly, 9.84% monthly, and 0.55% annually. Within the group using ectoparasiticides, a majority (39.89%) could not identify the medication used. Amitraz, afoxolaner, and fipronil were mentioned by 38.80%, 6.01%, and 4.92% of the owners, respectively.

Out of the 183 evaluated animals, 91.80% (168/183) had never been taken to veterinary consultations, 3.83% (7/183) were seen annually, 2.73% (5/183) semiannually, and 1.64% (3/183) monthly.

The owners’ educational levels were assessed during interviews. The majority (36.07%, 22/61) had completed high school, 32.79% (20/61) had a college degree or technical education, and 31.15% (19/61) were either illiterate or had only completed elementary school. Although 88.52% (54/61) were unfamiliar with the term “zoonoses,” most of the dog owners (83.61%, 51/61) were aware of the cross-transmission of diseases between dogs and humans. Rabies (32.79%, 20/61), scabies (9.84%, 6/62), leptospirosis, and leishmaniosis (both 4.92%, 3/61) were the diseases mentioned by those aware of the risk of disease transmission. Regarding parasitic diseases, most of the owners mentioned ticks (98.36%), cutaneous larva migrans (96.72%), toxoplasmosis (70.49%), and giardiosis (68.85%). The least known parasites were *Cryptosporidium* (8.20%), *Cystoisospora* (6.56%), and *Toxocara canis* (4.92%). Awareness of zoonoses was associated with the level of education (refer to Table 5); however, the recognition of cross-transmission of diseases between dogs and humans remained consistent across all educational categories (*p* = 1).

## 4. Discussion

This study encountered limitations, including the relatively basic laboratory facilities in Rondônia, limited personnel availability, and budgetary constraints. Despite these limitations, it is important to highlight that this study contributes to the understanding of parasitic infections in pet dogs in the state of Rondônia, which remains limited due to the scarcity of research on canine diseases in the northern region of Brazil. Our findings revealed a significant proportion of infected pet dogs in the municipality of Rolim de Moura, underscoring the necessity for further investigation given the substantial sample size. Previous studies showed a high prevalence of gastrointestinal parasite infections in pet dogs in two cities within the state, where infection rates reached 84.2% (72/95) and 87.5% (35/40) [9,10]. Correspondingly, similar research conducted in other states of the northern region also reported elevated infection rates. For instance, a study that evaluated 80 stray dogs in Manaus, Amazonas, revealed positive parasite findings in all animals [11]. Furthermore, another study conducted in Gurupi, Tocantins, exhibited a gastrointestinal parasite infection rate of 39.20% among 126 stray dogs [2].

Ensuring the overall well-being of pet dogs, including their health, requires responsible ownership and proper care. Regular veterinary attention plays a crucial role in this context. However, it is concerning that only 8.20% of the animals in this study received regular veterinary attention. Furthermore, most dog owners incorrectly administered both anthelmintics and ectoparasiticides. According to the guidelines provided by the European Scientific Counsel Companion Animal Parasites and the Tropical Council for Companion Animal Parasites [24], anthelmintics should be administered at least quarterly, while ectoparasiticides should be administered monthly. The misuse of these medications can be attributed to a lack of knowledge and guidance among dog owners regarding responsible pet ownership. Pet owners bear the responsibility of meeting the physical, psychological, and environmental needs of their pets, while also taking measures to prevent risks, such as aggression, disease transmission, and harm to others [25].

The dynamics of certain tropical parasitic diseases have witnessed global changes, which have been influenced by socio-environmental factors [26]. In the Amazon region, the conversion of native forests to other land uses carries significant consequences. Newly populated areas often lack proper basic sanitation infrastructure, leading to the spread of zoonotic helminth infections transmitted through the soil [27]. Despite boasting a relatively high human development index (0.700) in comparison with other cities in Rondônia, only 14.6% of households in Rolim de Moura have access to a suitable water supply and sewage collection [15]. Most homes rely on septic tanks and wells for their water needs. In municipalities lacking adequate sewerage systems, groundwater can become contaminated due to improper septic tank waste disposal or irregular sewage, consequently resulting in environmental contamination [28]. Given the prevalence of zoonotic parasites in this area, this has additional negative impacts on the health of the population, further exacerbating diseases associated with inadequate sanitation. Engaging in activities focused on human and animal health could yield substantial benefits for the local population, including effectively controlling diagnosed parasites and enhancing sanitary conditions for both dogs and humans. 

Two important zoonotic parasites, namely, *Ancylostoma* spp. and *Toxocara canis*, were found to be prevalent in Rolim de Moura. However, no available data pertains to human infections caused by these parasites in the city. The risk of exposure to these diseases extended beyond the dog owners at home, as 41.53% of dogs had outdoor access. Consequently, other individuals can potentially be exposed to these parasites when the dogs roam the streets during walks in public areas and parks [2]. This inadequate interaction between humans, animals, and the environment heightens the risk of zoonoses, particularly among children who engage in outdoor activities and potential geophagy, rendering them more susceptible to these pathogens [29]. The larval form of *Ancylostoma* spp., which was identified in 68.71% of the dog samples in this study, is the causative agent of cutaneous larva migrans (CLM) disease. CLM is endemic in tropical and subtropical regions, particularly in Latin America and the Caribbean, Southeast Asia, and Africa. Accurate assessments of the global occurrence of CLM remain lacking. Publication bias might be present due to underreporting and potential underdiagnosis in numerous countries where this condition is often overlooked [30], which also happens in northern Brazil. 

Infection by *T. canis* was observed in 6.75% of the tested dogs. This apparent low frequency may be related to the parasite’s biology, as in dogs older than four to six months, the parasite’s larvae tend to enter a hypobiotic state in the host’s tissues instead of developing into the adult stage [31]. Most sampled dogs were over one year old, suggesting that the observed prevalence might not accurately reflect the true distribution of this nematode. Toxocariosis, which is caused by the nematode *T. canis*, involves humans as paratenic hosts, harboring larvae of the parasite in their tissues [32]. Presently, toxocariosis ranks among the six most significant neglected parasitic infections in the United States [33]. Apart from ocular damage, human toxocariosis is associated with neurological lesions, leading to cognitive deficits in exposed children [34]. There is inadequate data on the occurrence of *T. canis* infection in dogs across various Brazilian regions, yet available information suggests that its prevalence could be higher in northern Brazil compared with other areas. Importantly, the burden of human toxocariosis in this country remains largely underestimated [35]. 

Concerning ectoparasites, *Rhipicephalus sanguineus* ticks were detected in the evaluated animals, particularly in dogs residing in urban areas [36]. Notably, purebred animals exhibited significantly fewer ectoparasites, possibly due to better care and the use of costly ectoparasiticides, which might be less accessible for mixed-breed dog owners. 

It is noteworthy to say that the present study is based on propagule morphology, and the employed methodology could underestimate the presence of parasites with small propagules, such as *Giardia* spp. and *Cryptosporidium* spp. The absence of these parasites in our results could be related to the low sensitivity of the adopted techniques. Considering *Giardia* spp., the combination of rapid tests and fecal flotation could reduce the chances of false negatives [37,38,39,40]. On the other hand, the diagnosis of *Cryptosporidium* spp. infection depends on specific techniques, such as acid-fast staining or molecular tests [41,42]. Also, as the study area is inserted in one of the most megadiverse regions of the planet [43], the occurrence of rare or scarcely known parasite species in the studied animals should not be ruled out, as recent studies developed in Australia, which is another megadiverse country, revealed the presence of *Linguatula serrata* affecting dogs and wild carnivores. As these parasites’ eggs resemble the propagules of other parasites, their presence was ignored [44,45]. Therefore, further studies should be encouraged in this region, adopting other techniques and sampling the wild animals associated with anthropized areas.

Campaigns that endorse responsible pet ownership and the prevention of zoonotic diseases can play a pivotal role in bridging this gap, as their scope extends beyond the financial aspect of pet ownership. Additionally, it is essential to underscore that alongside responsible ownership campaigns, health education assumes a pivotal role in mitigating concerns related to animal and public health. Employing questionnaires to gauge the population’s existing knowledge of zoonoses allows for the formulation of tailored health education policies, which can be developed based on the unique requirements of each community. These initiatives can foster collaborative efforts in health education and epidemiological surveillance [46].

## 5. Conclusions

The prevalence of parasite infection, particularly those with zoonotic potential, among pet dogs was high in Rolim de Moura. The close interaction between humans and infected animals highlights the significance of addressing these potential health risks. Implementing effective educational initiatives can play a crucial role in reducing the incidences of these zoonotic diseases and promoting a safer environment within the municipality.

## Figures and Tables

**Figure 1 ijerph-21-00138-f001:**
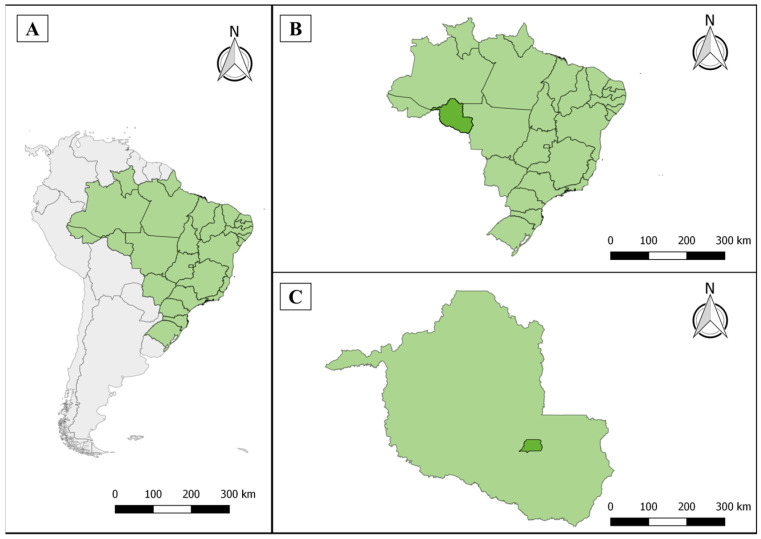
Study area. South America, highlighting Brazil (**A**). State of Rondônia (**B**). Municipality of Rolim de Moura (**C**), located in the Zona da Mata region.

**Figure 2 ijerph-21-00138-f002:**
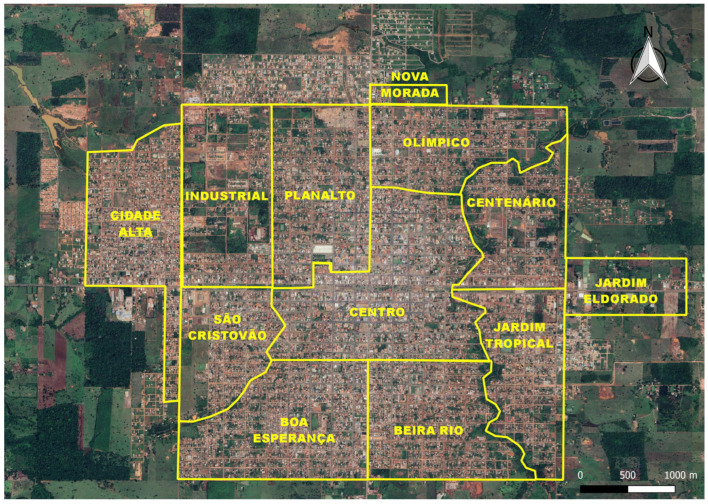
Municipality of Rolim de Moura, Rondônia State, Brazil, highlighting the 12 sectors where the study was conducted: Beira Rio, Boa Esperança, Centenário, Centro, Cidade Alta, Industrial, Jardim Eldorado, Jardim Tropical, Nova Morada, Olímpico, Planalto, and São Cristóvão.

**Figure 3 ijerph-21-00138-f003:**
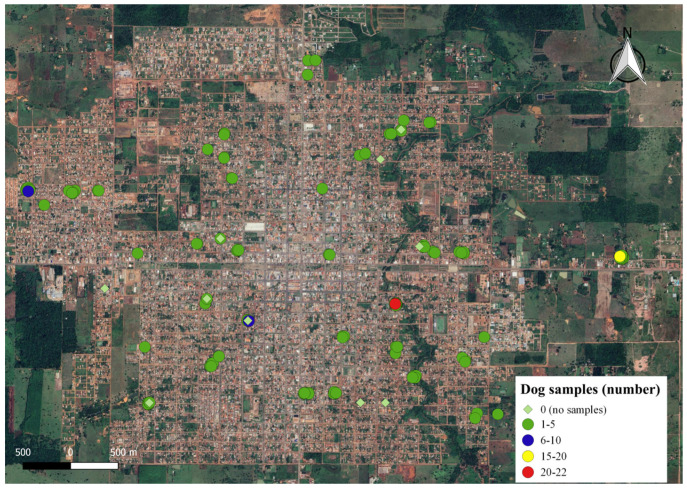
Number of fecal samples collected per household in the Municipality of Rolim de Moura, Rondônia State, Brazil.

**Figure 4 ijerph-21-00138-f004:**
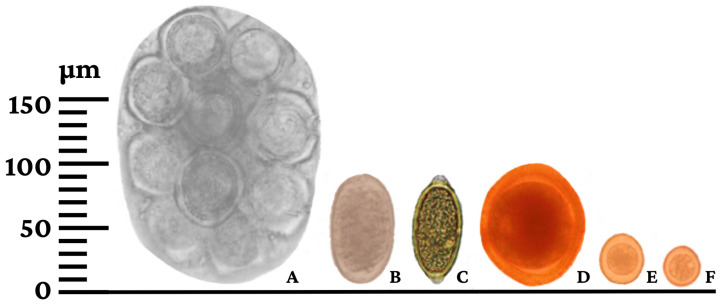
Helminth eggs and protozoan oocysts diagnosed in pet dogs from Rolim de Moura, Rondônia state, Brazil. (**A**) *Dipylidium caninum*; (**B**) *Ancylostoma* spp.; (**C**) *Trichuris vulpis*; (**D**) *Toxocara canis*; (**E**) *Cystoisospora canis*; (**F***) Hammondia*/*Neospora*.

**Table 1 ijerph-21-00138-t001:** Gastrointestinal parasites in pet dogs from Rolim de Moura, Rondônia state, Brazil, 2019.

Gastrointestinal Parasites		
Species	Prevalence	95% CI (%)
*Ancylostoma* spp.	68.71% (112/163)	60.99–75.74
*Trichuris vulpis*	11.66% (19/163)	7.17–17.60
*Toxocara canis*	6.75% (11/163)	3.42–11.75
*Cystoisospora canis*	4.91% (8/163)	2.14–9.44
*Dipylidium caninum*	1.23% (2/163)	0.15–4.36
*Hammondia/Neospora*	0.61% (1/163)	0.02–3.37
*Ancylostoma* spp. and *Trichuris vulpis*	8.59% (14/163)	4.78–13.99
*Ancylostoma* spp. and *Toxocara canis*	4.91% (8/163)	2.14–9.44
*Ancylostoma* spp., *Toxocara canis*, *Cystoisospora canis*, and *Hammondia Neospora*	0.61% (1/163)	0.02–3.37
Total	74.23% (121/163)	68.81–80.76

**Table 2 ijerph-21-00138-t002:** Descriptors of endoparasitic infection in pet dogs from Rolim de Moura, Rondônia State, Brazil, 2019, grouped in city sectors.

Sectors	P (inf/n)	CI (95%)	MI ± SD	RI
Beira Rio				
*Ancylostoma* spp.	41.66% (5/12)	15.17–72.33	250 ± 566.49	100–1900
*Trichuris vulpis*	0% (0/12)	-	-	-
*Toxocara canis*	0% (0/12)	-	-	-
Total	41.66% (5/12)	15.17–72.33	250 ± 566.49	100–1900
Boa Esperança				
*Ancylostoma* spp.	88.24% (15/17)	65.56–98.54	1126.67 ± 1084.82	100–3000
*Trichuris vulpis*	17.65% (3/17)	3.80–43.43	433.33 ± 201.65	100–700
*Toxocara canis*	5.88% (1/17)	0.15–28.69	3100 ± 751.87	-
Total	88.24% (15/17)	65.56–98.54	1252.94 ± 1307.73	200–3700
Centenário				
*Ancylostoma* spp.	33.33% (4/12)	9.92–65.11	625 ± 464.09	200–1600
*Trichuris vulpis*	0% (0/12)	-	0	-
*Toxocara canis*	8.33% (1/12)	0.21–38.48	200 ± 57.74	-
Total	33.33% (4/12)	9.92–65.11	675 ± 467.34	200–1600
Centro				
*Ancylostoma* spp.	75.68% (28/37)	58.80–88.23	1508.93 ± 1630.04	50–5700
*Trichuris vulpis*	10.81% (4/37)	3.03–25.42	475.00 ± 156.56	50–750
*Toxocara canis*	5.41% (2/37)	0.66–18.19	350.00 ± 87.68	200–500
Total	83.78% (31/37)	67.99–93.81	1446.77 ± 1678.85	50–7100
Cidade Alta				
*Ancylostoma* spp.	37.5% (6/16)	15.20–6457	833.33 ± 865.54	100–3500
*Trichuris vulpis*	31.25% (5/16)	11.02–58.66	160.00 ± 89.44	100–300
*Toxocara canis*	12.5% (2/16)	1.55–38.35	2000.00 ±707.11	1500–2500
Total	68.75% (11/16)	41.34–88.98	890.91 ± 1024.29	100–3500
Jardim Eldorado				
*Ancylostoma* spp.	75% (15/20)	50.90–91.34	7700.00 ± 9561.21	100–39,000
*Trichuris vulpis*	0% (0/20)	-	-	-
*Toxocara canis*	5% (1/20)	0.13–28.87	1700.00 ± 380.13	-
Total	80% (16/20)	56.34–92.27	7325.00 ± 9514.61	100–39,000
Jardim Tropical				
*Ancylostoma* spp.	70.59% (12/17)	44.04–89.69	4691.67 ± 11,150.95	100–46,100
*Trichuris vulpis*	0% (0/17)	-	-	-
*Toxocara canis*	11.76% (2/17)	1.46–36.44	2250.00 ± 1016.70	200–4200
Total	70.59% (12/17)	44.04–89.69	5066.67 ± 12,154.19	100–50,300
Nova Morada				
*Ancylostoma* spp.	100% (3/3)	29.24–100	600 ± 700.00	100–1400
*Trichuris vulpis*	0% (0/3)	-	-	-
*Toxocara canis*	0% (0/3)	-	-	-
Total	100% (3/3)	29.24–100	600 ± 700.00	100–1400
Olímpico				
*Ancylostoma* spp.	80% (8/10)	44.39–97.48	262.5 ± 152.39	100–400
*Trichuris vulpis*	20% (2/10)	2.52–55.61	350 ± 149.44	300–400
*Toxocara canis*	0% (0/10)	-	-	-
Total	80% (8/10)	44.39–97.48	350 ± 261.62	100–800
Planalto				
*Ancylostoma* spp.	41.67% (5/12)	15.17–72.33	1100.00 ± 712.82	500–2300
*Trichuris vulpis*	0% (0/12)	-	-	-
*Toxocara canis*	0% (0/12)	-	-	-
Total	41.67% (5/12)	15.17–72.33	1100.00 ± 712.82	500–2300
São Cristovão				
*Ancylostoma* spp.	66.67% (4/6)	22.28–95.67	2150 ± 1590.81	700–4200
*Trichuris vulpis*	0% (0/6)	-	-	-
*Toxocara canis*	0% (0/6)	-	-	-
Total	66.67% (4/6)	22.28–95.67	2150 ± 1590.81	700–4200

n—number of dogs examined; POS—number of dogs infected; P—prevalence; (inf./n)—proportion of infected dogs in the sample; CI (95%)—confidence interval; MI ± SD—mean intensity ± standard deviation; RI—range of intensity (eggs per gram of feces). Note: the industrial sector was omitted due to the inclusion of only one sampled animal.

**Table 3 ijerph-21-00138-t003:** Univariate and multivariate analysis of the association between dog characteristics, environmental characteristics, and frequency of parasitic infection in pet dogs from Rolim de Moura, Rondônia State, Brazil, 2019.

Univariate Statistical Analysis					
Variables	n	INF	%	(95% CI)	*p*
Sex					
Male	58	46	79.31	66.65–88.83	0.35
Female	105	75	71.43	61.79–79.82	
Size					
Small	81	57	70.37	59.19–80.01	0.53
Medium	48	38	79.17	65.01–89.53	
Large	34	26	76.47	58.83–89.25	
Breed					
Purebred	41	25	60.98	44.50–75.80	0.03
Mixed breed	122	96	78.69	70.35–85.58	
Age					
Up to 12 months	37	31	83.78	67.99–93.81	0.13
Over 12 months	126	90	71.43	62.70–79.12	
Outdoor access (walks)					
Yes	60	46	76.67	63.96–86.62	0.71
No	103	75	72.82	63.16–81.12	
Use of antihelminthics					
Yes	123	88	71.54	62.71–79.31	0.21
No	40	33	82.50	67.22–92.66	
Residence location					
Near the river	30	23	76.67	57.72–90.07	0.30
Away from the river	27	24	88.89	70.84–97.65	
Street paving					
Asphalted	24	21	87.50	67.64–97.34	0.49
Not asphalted	33	26	78.79	61.09–91.02	
Sewage system					
Yes	5	4	80.00	28.36–99.49	1.00
No (septic tank)	52	43	82.69	69.67–91.77	
Water source					
Treated	6	5	83.33	35.88–99.58	0.87
Untreated (well)	16	14	87.50	61.65–98.45	
Both	35	28	80.00	63.06–91.56	
Frequency of veterinary consultations					
Never	149	116	77.85	70.33–84.24	0.007
Yearly	7	3	42.86	9.90–81.59	
Semiannually	4	0	0	0.00–60.25	
Monthly/quarterly	3	2	66.67	9.43–99.16	
House access					
Indoor access	21	10	47.62	25.71–70.22	0.004
Outdoor only	142	111	78.17	70.47–84.66	
Multivariate analysis					
Variables		Odds ratio	(95% CI)	*p*	
Frequency of veterinary consultations		0.80	0.61–0.95	0.025	
Indoor access		0.23	0.06–0.84	0.020	

n—number of dogs examined; INF—number of infected dogs; %—frequency; 95% CI—95% confidence interval; *p*—*p*-value.

**Table 4 ijerph-21-00138-t004:** Univariate analysis of the association between in pet dogs characteristics and frequency of infection by *Ancylostoma* spp., Rolim de Moura, Rondônia State, Brazil, 2019.

Variables	n	INF	%	(95% CI)	*p*
Sex					
Male	58	42	66.67	59.10–83.34	0.48
Female	105	70	72.41	56.80–75.57	
Size					
Small	81	53	65.43	54.04–75.66	0.66
Medium	48	35	72.92	58.15–84.72	
Large	34	24	70.59	52.52–84.90	
Breed					
Purebred	41	24	58.54	42.11–73.68	0.11
Mixed breed	122	88	72.13	63.29–79.87	
Age					
Up to 12 months	37	29	78.38	61.79–90.17	0.16
Over 12 months	126	83	65.87	56.90–74.08	
Outdoor access (walks)					
Yes	60	45	75.00	62.14–85.28	0.22
No	103	67	65.05	55.02–74.18	
Use of antihelminthics					
Yes	123	81	65.85	56.76–74.16	0.23
No	40	31	77.50	61.55–89.16	
Frequency of veterinary consultations					
Never	149	107	71.81	63.87–78.87	0.02
Yearly	7	3	42.86	9.90–81.59	
Semiannually	4	0	0	0.00–60.24	
Monthly/quarterly	3	2	66.67	9.43–99.16	

n—number of dogs examined; INF—number of infected dogs; %—frequency; 95% CI—95% confidence interval; *p*—*p*-value.

**Table 5 ijerph-21-00138-t005:** Analysis of the association between the level of education and knowledge of the dog owner about the concept of zoonoses, Rolim de Moura, Rondônia State, Brazil, 2019.

Education Level	n	Knowledge of the Concept of Zoonoses		Logistic Regression	
		Yes	% (95% CI)	*p*	OR (CI 95%)
Illiterate/elementary school *	19	0	0.00 (0.00–16.82)	0.0223	10.75 (1.40–82.40)
High school	22	1	4.55 (0.81–21.80)		
Undergraduate/graduate/technical	20	6	30.00 (14.55–51.90)		
Total	61	7	11.48 (5.68–21.84)		

*—reference level; n—number of dog owners interviewed; % (95% CI)—frequency and 95% frequency confidence interval; *p*—*p*-value; OR (CI 95%)—odds ratio and its 95% confidence interval.

## Data Availability

The data presented in this study is contained within the article.

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
