# Peer review of "Parasitosis in Pet Dogs from Rondônia, Amazon Biome, and Human Perception of Zoonoses"

_ijerph, 2024, doi:10.3390/ijerph21020138_

Round 1
Reviewer 1 Report (Previous Reviewer 1)
Comments and Suggestions for Authors
Dear Authors,
sorry for the delay of my report
I read the manuscript and I found it improved, especially because you stated clearly what are the limits of your work
You answered at almost all my comments. Nevertheless I still have found something that should be emended, according to my opinion.
In particular, I think that you should refer the laboratory methods to any international recognized manual (Urquarth at al; Taylor et al; Zajac), instead of old and regional texts you mentioned.
I attach the manuscript with some others comments by my side
Best regards

Author Response
Please see the attachment.

Reviewer 2 Report (New Reviewer)
Comments and Suggestions for Authors
Congratulations, the manuscript are very interesting because the relation between human and animals like pets should be evaluated to modulate immunity specially in children. Other hand will be deleterious in case of zoonoses and because of that fact should be promote in the community, school activities to improve sensibilization about the risks and benefits of pets. Aditionally the impact of ancylostoma may vary on anemia, death on human immunosupressed and allergic disease in case of toxocara transmission.In your study, did not found any strongyloides stercoralis infection?Why?
Greatful to reading aal the manuscript but particularly the discussion...in my oppinion it was well described the limitations, methodologies (isolated and in combination) and results.
Author Response
Please see the attachment.

This manuscript is a resubmission of an earlier submission. The following is a list of the peer review reports and author responses from that submission.
Round 1
Reviewer 1 Report
Comments and Suggestions for Authors
The manuscript "Parasitosis in pet dogs from Rondônia, Amazon Biome: risk factors and human perception of zoonoses" reported the results of a local study on parasitosis of Dogs and risk perception by owners.
The question posed by the Authors is correct (how much are dogs parasitized, by what parasites and are the owners aware of risk for their health and wellbeing?).
The main constraint of the study is the very basic methodology applied both to the study design and to the laboratory methods used. Vice versa, the statistical method, even if simple, results appropriate.
Regarding the first, there was no one effort to design a representative sampling. Regarding the laboratory procedures, even if I have no access to the cited literature, I guess just a simple floatation was used. This is absolutely not sufficient to detect some parasites of great importance in public health (i.e.Echinococcus sp, Giardia sp.).
Moreover, specific commets are reported in the attched copy of Ms.
I suggest to the Authors to submit the paper to a local Journal on Veterinary or public health.
Best regards

It needs language editing
Reviewer 2 Report
Comments and Suggestions for Authors
The authors aim to identify the ocurrence of parasitosis in pet dogs in a specific part of Rondônia state, evaluate risk factors associated with the animal-human-environment interface, and evaluate the tutor perception of parasites and zoonoses.
In general terms, the manuscript is well-written. This is an issue of great importance, with a One Health vision, but the manuscript lacks a proper research method, there is no info on why only two regression models are presented for parasitic species, and the zoonoses knowledge is not presented in the form of multivariable regression. Further details are needed in the M6M and Results sections, both will strengthen the discussion of this work.
Specific comments/suggestions are highlighted in the attached .pdf file.

Reviewer 3 Report
Comments and Suggestions for Authors
Dear Author thank you for submitting manuscript entitle "Parasitosis in pet dogs from Rondônia, Amazon Biome: risk factors and human perception of zoonoses"
Article is well written; selection, and conduct of study is also good. Sample size and area coverage is good, geographical presentation is impressive.
Please address following queries:
1) Please briefly describe the method of fecal processing. (Fecal flotation/Sheather's solution/ or any method related to processing).
2) Please justify risk factor in title by doing risk factor analysis or else remove it from the title.
3) Please avoid writing species level name for any complicated parasite that is complicated to identify through microscope.
4) please attach original photograph of Helminth eggs and protozoan oocysts as supplymentary file or directly present them in manuscript. Figure 4 is not good way of presenting a study.
5) Dogs carrying Dipylidium caninum; Ancylostoma spp.; Toxocara canis, were ill in health or good in health? and what was the egg quantity from that perticular cases?
Comments on the Quality of English LanguageMinor changes
Round 2
Reviewer 1 Report
Comments and Suggestions for Authors
Dear Authors,
Thank you for your revision. I appreciate the effort made to improve the manuscript after the first round of reviewing.
Unfortunately, some important criticisms are still in place. First of all, the dog population sampled. The absolute number could be adequate, but I do not agree with the choice to sample more than one dog in the same household. It does not add any epidemiological values and impair the representativeness of the sample. There are also issues with the laboratory methods, not able to detect Giardia and to identify some helminths. The reference section is improved, but the numeration of references on the lab methods is wrong (items 22-25, does not match with the citations). Even the caption of the Fig. 1 is wrong (where is the "D" ?).
In general, I think that the manuscript is not of the quality level requested by IJEHRP, therefore my recommendation is to reject the paper. Anyway I would recommend to the Authors to resubmit it to an other journal with a lower profile, and on which the interesting results obtained could be published.
Best regards
Author Response
|
Response to Reviewer 1 Comments
|
|
Thank you very much for taking the time to review this manuscript. Please find the detailed responses below and the corresponding revisions/corrections highlighted in purple in the re-submitted files.
|
|
1. Point-by-point response to Comments and Suggestions for Authors |
|
Comment 1: Thank you for your revision. I appreciate the effort made to improve the manuscript after the first round of reviewing. Unfortunately, some important criticisms are still in place. First of all, the dog population sampled. The absolute number could be adequate, but I do not agree with the choice to sample more than one dog in the same household. It does not add any epidemiological values and impair the representativeness of the sample. There are also issues with the laboratory methods, not able to detect Giardia and to identify some helminths.
In general, I think that the manuscript is not of the quality level requested by IJEHRP, therefore my recommendation is to reject the paper. Anyway I would recommend to the Authors to resubmit it to an other journal with a lower profile, and on which the interesting results obtained could be published.
Response 1: We respectfully disagree, as most dog parasites have aggregated distribution and, even in the same households, host characteristics such as age or health status could influence on the parasite status. Even though other methods and rapid tests could be used to improve the diagnostic sensibility, the results have merit and should be considered, as even giardosis could be diagnosed in cases with moderate to high cyst counts. We could even detect oocysts of Hammondia/Neospora, indicating the techniques were well executed.
Comment 2: “The reference section is improved, but the numeration of references on the lab methods is wrong (items 22-25, does not match with the citations).”
|
Response 2: Done.
Comments 3: “Even the caption of the Fig. 1 is wrong (where is the "D" ?).
Response 3: Done.